# Posterior Tibial Nerve Stimulation in Children with Lower Urinary Tract Dysfunction: A Mixed-Methods Analysis of Experiences, Quality of Life and Treatment Effect

**DOI:** 10.3390/ijerph19159062

**Published:** 2022-07-25

**Authors:** Liesbeth L. De Wall, Anna P. Bekker, Loes Oomen, Vera A. C. T. Janssen, Barbara B. M. Kortmann, John P. F. A. Heesakkers, Anke J. M. Oerlemans

**Affiliations:** 1Division of Pediatric Urology, Radboud University Medical Center, Amalia Children’s Hospital, 6525 GA Nijmegen, The Netherlands; pluk.bekker@ru.nl (A.P.B.); loes.oomen@radboudumc.nl (L.O.); vera.act.janssen@radboudumc.nl (V.A.C.T.J.); barbara.kortmann@radboudumc.nl (B.B.M.K.); 2Department of Urology, University Hospital Maastricht, 6229 HX Maastricht, The Netherlands; john.heesakkers@mumc.nl; 3Department of IQ Healthcare, Radboud University Medical Center, 6525 GA Nijmegen, The Netherlands; anke.oerlemans@radboudumc.nl

**Keywords:** PTNS, children, quality of life, urinary incontinence, qualitative outcomes

## Abstract

Background: Posterior tibial nerve stimulation (PTNS) is one of the treatment modalities for children with therapy-refractory lower urinary tract dysfunction (LUTD). This study used a mixed-methods analysis to gain insight into the experiences of children treated with PTNS and their parents, the effect of treatment on quality of life (QOL) and the effect of PTNS on urinary symptoms. Methods: Quantitative outcomes were assessed through a single-centre retrospective chart analysis of all children treated with PTNS in a group setting between 2016–2021. Voiding parameters and QOL scores before and after treatment were compared. Qualitative outcomes were assessed by an explorative study involving semi-structured interviews transcribed verbatim and inductively analysed using the constant-comparative method. Results: The data of 101 children treated with PTNS were analysed. Overall improvement of LUTD was seen in 42% and complete resolution in 10%. Average and maximum voided volumes significantly increased. QOL improved in both parents and children independent of the actual effect on urinary symptoms. Interviews revealed PTNS to be well-tolerated. Facilitating PTNS in a group setting led to feelings of recognition in both children and parents. Conclusions: PTNS is a good treatment in children with therapy-refractory LUTD and provides valuable opportunities for peer support if given in a group setting.

## 1. Introduction

Lower urinary tract dysfunction (LUTD) is a common condition, with a prevalence up to 21% in otherwise healthy school aged children [1,2]. It is associated with a negative impact on quality of life (QOL), lower self-esteem, social stigmatization and impaired interpersonal interactions [3,4]. Besides, LUTD can have a major impact on a family’s wellbeing. Whereas some parents succeed in adapting their daily routines to minimize the impact of the child’s condition, other parents experience high degrees of stress, struggle and become frustrated as they try to adjust to their child’s wetting. With subsequent, numerous and ineffective treatments, parents lose hope and optimism [5].

Common treatments such as urotherapy, cognitive behavioral treatment, biofeedback training, pelvic floor treatment and medication to suppress bladder overactivity (antimuscarinics) fail in approximately 20–40% of these children [6]. In addition, efficacy of antimuscarinics is limited by the known and unwanted side effects, such as behavioral changes and constipation. Compliance is low and many patients discontinue antimuscarinics in the long term (88% at four years) [7]. Not surprisingly, parents are reluctant to give medication for a long period to their otherwise healthy children.

Second-line treatment options for children with therapy-resistant urgeurinary incontinence include other types of (mainly off-label) medication, intravesical botulinum toxin A injections or neuromodulation [6]. In the latter, stimulation of specific peripheral nerves or their dermatomes intend to cause an alteration of the afferent and efferent neurological pathways between the brain, brain stem and pelvic organs involved in the micturition cycle, with subsequent normalization of abnormal function of the bladder [8]. Posterior tibial nerve stimulation (PTNS) is one of the forms of neuromodulation given in children with a response varying between 31–78% [9,10,11]. Despite its potential positive effect on urinary symptoms, experiences of children and their parents with this time-consuming treatment and the effect on QOL are not well-known.

The aim of this study is to gain insight into the experiences of children treated with PTNS and their parents, the effect of treatment on QOL and the effect of PTNS on urinary symptoms. More knowledge of the effect of PTNS on QOL and how PTNS is experienced, with the help of qualitative research, can help to establish optimal treatment protocols in the future.

## 2. Materials and Methods

This is a mixed-methods study including both qualitative and quantitative research methods. The quantitative outcomes analysed are treatment response, change in frequency voiding chart (FVC) parameters and change of QOL. The qualitative outcomes are the experiences of children and their parents with PTNS.

### 2.1. PTNS Treatment

PTNS treatment included 30 min of stimulation once weekly for 12 consecutive weeks at the outpatient clinic using a standard device (Urgent PC^®^; Cogentix Medical Inc., Minnetonka, MN, USA). Stimulation (200 usec, 20 Hz frequency and titrated at 0–10 mA) was given via a 34-gauge needle inserted two fingers to the medial malleolus of the ankle [12] (Figure 1). Proper needle placement was confirmed by observing ipsilateral plantar and/or toe flexion. In all patients, a topical anesthetic agent (lidocaine cream) was used to reduce pain and fear associated with needle insertion, which was applied by parents an hour before treatment. Needle insertion was performed by a specialized, trained pediatric nurse. A psychological co-worker was present to give mental support, if needed, during needle insertion. All children received their treatment in a group setting, changing from 3–6 patients at a time varying in age, gender and status of their current PTNS treatment (at the beginning-halfway-end). Use of a mobile device during treatment was allowed and contact with other children/parents was stimulated. After the 12th PTNS session, the treatment was evaluated by the physician. Those with a complete or partial response on urinary symptoms were offered a tapering schedule.

### 2.2. Quantitative outcome: Treatment Response and Frequency Voiding Chart Parameters

Inclusion criteria were all children aged 6–12 with LUTD, treated in our tertiary clinic with PTNS. This included a total of 101 children whom were analysed retrospectively. Neurogenic lower urinary tract dysfunction was an exclusion criterion. LUTD was defined according to the International Children’s Continence Society (ICCS) [13]. Treatment response and change in frequency voiding chart parameters before and after treatment were assessed. FVC parameters included average and maximum voided volumes [13]. Treatment response was assessed by change in urinary symptoms quantified according to Mulders et al. and further classified according to international guidelines in: complete response (100% resolution of symptoms), partial response (50–99% reduction in symptoms) or no response (<50% reduction in symptoms) [13,14].

### 2.3. Quantitative Outcome: Quality of Life

Since 2019, disease-specific QOL assessment became part of standardized treatment in our PTNS protocol. The validated Pediatric Incontinence Questionnaire (PIN-Q) for both parents and child was filled out before PTNS and at the end of treatment after the 12th session. The PIN-Q is a 20-item disease-specific QOL questionnaire scored on a 5-point Likert scale, with a higher total score indicating a lower QOL. Thibodeau et al. made an assumption to grade the severity of impact on Quality of life based on the PIN-Q total score: mild <20, moderate 21–50, severe >51 [5].

### 2.4. Statistical Analysis of Quantitative Outcomes

Treatment outcome and frequency voiding chart parameters were expressed as numbers and proportions (mean ± standard deviation [SD] or median and 25th–75th quartiles). A paired t-test was performed for normally distributed continuous data, and a Mann–Whitney U test was used for non-normally distributed continuous data. The data were analysed using SPSS Statistics 25.0 (SPSS Inc., Chicago, IL, USA). Differences were considered statistically significant at *p* < 0.05.

### 2.5. Qualitative Outcome: Experiences of Children and/or Parents with PTNS

To study experiences of children and parents, a subsequent explorative qualitative study was done in a subgroup using semi-structured in-depth interviews. Interviews were done with parents alone or parents and their child, depending on the child’s preference. An interview design was chosen, as actual experiences are not easily measured by questionnaires or interpreted through observation. Individual in-depth interviews were chosen instead of group interviews as LUTD and associated psychological impact is considered a sensitive topic.

Children receiving PTNS at the time of the study, or those who had completed PTNS in the past 18 months, as well as their parents, were invited. A threshold of 18 months was chosen to minimalize the risk of recall bias and/or loss of memories of experiences due to additional treatments following PTNS. To ensure a varied group of children and parents, a purposive sampling strategy was used to select the children regarding the following characteristics: gender, age, overall duration of treatment before PTNS and current duration of PTNS (in those receiving PTNS at the time of the study). Parents and children were provided with study information either before or after a PTNS session or by email and subsequently invited to participate. Written informed consent was obtained in all parents and children ≥12 years of age.

A topic guide was designed based on the available literature and clinical experiences from doctors, nurse practitioners and physician assistants involved in treating children with LUTD (Appendix A). After the first interviews, the topic guide was slightly adjusted to better fit our research question. All interviews took place between October and November 2021. The interviews (45–60 min) were performed in person before or after a PTNS session or online, using video-conferencing software. Recruitment and interviewing proceeded until data saturation was reached, and no new findings emerged from subsequent interviews.

Interviews were conducted by an independent research member, trained in qualitative research and interviewing, who had no professional relationship with parents and/or child (P.B.). Interviews were audio-recorded, transcribed verbatim, anonymized and subsequently analysed using ATLAS.ti (version 9.1.6 Scientific Software Development, GmbH, Berlin, Germany). Field notes were made during the interviews. Inductive analysis was done, a qualitative approach focusing on identification of themes and concepts without predefined hypothesis [15]. All transcripts were analysed independently by analysts (P.B., L.L.D.W.). Any discrepancies were discussed until consensus was reached. The codes emerged from the data and were refined in an iterative process of coding, comparing and refining. They were subsequently grouped into themes by three members of the research team (P.B., L.L.D.W., A.O.). Quotes presented to illustrate the results were translated into English by an independent native speaker.

### 2.6. Ethical Considerations

The study was approved by the Research Ethics Committee Arnhem-Nijmegen (registration number: 2021-13133) and registered in an accessible study registry (ISRCTN68115364). Conduction and reporting of this study followed the STROBE criteria (quantitative part) and COREQ criteria (qualitative part), [16,17].

## 3. Results

### 3.1. Quantitative Outcomes

#### 3.1.1. Treatment Response and Frequency Voiding Chart Parameters

Data of 101 children treated with PTNS between March 2016 till December 2021, were analysed and included for analysis. A total of 59% boys (N = 60) and 41% girls (N = 41) were included. Mean age at time of first PTNS treatment was 9.7 ± 2.4 years. Patients were offered PTNS after a median of 1.5 years (0.5 – 3.5 years) of previous treatments. Previous treatments included; urotherapy (95%), antimuscarinics (88%), cystoscopy to rule out infravesical obstruction (61%) and pelvic floor therapy (50%). Overactive bladder with urinary incontinence was diagnosed according to the ICCS in 78% (N = 79). Remaining diagnoses were overactive bladder without incontinence (N = 5), non-monosymptomatic enuresis (N = 11) and dysfunctional voiding (N = 5). In Table 1, FVC parameters and treatment outcome according to ICCS criteria are shown. Treatment outcome was known in all, and missing FVC parameters at start or after PTNS was seen in only 8% (N = 8). The remaining 92% (N = 93) were considered a good representative for children with therapy-refractory LUTD treated in a third-line clinic. Both average voided volumes and maximum voided volumes increased significantly throughout PTNS treatment, 25 mL and 13 mL, respectively, *p =* 0.000, *p* = 0.025. In 10% (N = 10), urinary symptoms resolved completely defined as complete response. In 32% (N = 32), a decrease in urinary symptoms of 50–99% was seen, with 9 children reporting a decrease of 80% or more. This was defined as partial response. No response in urinary symptoms was seen in 58% (N = 59).

#### 3.1.2. Quality of Life

PIN-Q outcomes of both children and parents were assessed in 45% (N = 46)., The remaining 55 children were treated before 2019 when the PIN-Q was not yet part of our protocol. Median PIN-Q score before and after PTNS treatment decreased significantly in both parents and children, as shown in Table 2. Median PIN-Q score at baseline and after PTNS treatment in children were 25 and 19 (*p* = 0.001), respectively, and median PIN-Q score at baseline and after PTNS treatment in parents were 26 and 22, respectively, (*p* = 0.001). PIN-Q scores were further subclassified according to treatment response (no response, partial response, complete response), as shown in Table 3. Differences in PIN-Q score before and after PTNS treatment did not differ between outcome for both parents and children, *p* = 0.512 and *p* = 0.051, respectively. Disease-specific QOL improved in both parents and children throughout PTNS treatment, regardless of the actual effect of PTNS on urinary symptoms.

### 3.2. Qualitative Outcomes

#### 3.2.1. Experiences of Children and/or Parents

In a subgroup of 11 patients, a subsequent explorative qualitative study was conducted with semi-structured in-depth interviews with parents and/or child, to study the experiences with PTNS treatment. A total of 12 parents were asked, and 11 were willing to participate. One mother refused participation, being too busy. Characteristics of the subgroup interviewed are shown in Table 4. Seven out of the eleven interviews were with the mother and child, the remaining with the mother alone. Four themes were derived through data analysis: “decision to choose PTNS treatment and expectations”, “time investment”, “practical aspects of PTNS” and “group setting/talking about LUTD”. Quotes of participants are used to illustrate participants’ experiences with PTNS.

#### 3.2.2. Decision to Choose PTNS and Expectations

Participants described different reasons to opt for PTNS. Alternative treatment options, such as medication and botulinum toxin A, had already been tried or were considered less attractive given the unwanted side effects of medication or the requirement of general anesthesia in botulinum toxin A.


*“All those other options were more surgery-like. I didn’t like that very much. The only thing I thought acceptable was an injection in the leg. We think that is better than an operation”.*
Child P5 (6 years)


*“We would have done anything to get her off that medication”.*
Mother P7

Some parents considered PTNS as more or less a last-resort treatment option and were willing to try anything to improve their child’s LUTD.


*“Yes, I assume everyone with the same problem as us, would try anything to solve the problem so I thought let’s give it a try and hopefully it will help”.*
Mother P10

Expectations of the PTNS treatment varied between parents; some had high hopes for an effect, while others were neutral and deliberately tried not to have high expectations because of failed previous treatments.


*“Well, to be honest I thought, I just assume it doesn’t help, that way it can only be better than expected. That was our starting point. So, yes, no harm, no foul”.*
Mother P11

#### 3.2.3. Time Investment

PTNS is known to be a time-consuming treatment, not only as it requires 12 subsequent weekly hospital visits but also as it is not facilitated in every clinic in the Netherlands. Parents mentioned the long travelling time for PTNS and the required logistic challenge at home or work to be able to come to the hospital every week.


*“[…] in the end it was quite tough. Every Friday. For half a year afterwards, [father] and I, would say, ‘ah we do not have to go to the hospital anymore’. Because we had to, say, leave between 12 and 12.30, eh, and you were back home at 4 or 5 o’clock. And you constantly got stuck in traffic”.*
Mother P9

The participating children themselves mentioned that they regretted having to leave school early and not being able to meet up with friends on the day of treatment.


*“He was really disappointed not being able to meet with friends that period as it was normally my day off”.*
Mother P10

Parents and children frequently mentioned PTNS at home and facilitation in more places, as suggestions for improvement.


*“As such, I do think that if, if it’s at home, it will make it easier. That we do not have to go to the hospital all the time and that won’t take as many hours of our time”.*
Child P8 (12 years)

#### 3.2.4. Practical Aspects of PTNS

Nervousness before the first PTNS session, and more specifically anxiousness for the needle insertion, was mentioned frequently by both parents and children. Distraction of the child with a book was viewed as helpful. Children described the actual needle insertion differently. Some felt hardly anything, some experienced it as bit painful and others as a strange feeling. In none of the children did the PTNS treatment have to be discontinued due to fear or pain.


*“When they put the needle in, it doesn’t hurt, but sometimes if they turn it on a bit too hard, it does hurt a little”.*
Child P4 (9 years)

After needle insertion, the rest of the PTNS session was experienced as quite comfortable. Children and parents started reading a book or using their tablet or phone. One parent said that her child was so at ease with the needle and the treatment that he wanted to disconnect the needle himself at the end of the session.


*“I think the first two or three times you were quite tense but now it’s like a walk in the park, isn’t it?”*
Mother P7


*“Sometimes it is quite relaxing, quite often actually because you can’t do anything at all; just read a book or something like that […]”.*
Child P4 (9 years)

#### 3.2.5. Group Setting/Talking about LUTD

In our centre, PTNS is given in a group setting of three–six children at a time, therefore, parents and children were specifically asked how they experienced the treatment in a group setting and about interaction with others regarding their LUTD. The interviews revealed different insights regarding openness in children concerning their urinary problems. Some kept it a secret from everyone else, whilst others shared it with the whole class in the form of a presentation.


*“My only friend is [name friend] and she won’t tell anyone. […] because I’m afraid that everyone will know and then they will laugh at me”.*
Child P5 (6 years)

Talking about LUTD and urinary incontinence during PTNS sessions differed as well among the interviewed subjects. Some parents mentioned dishonesty in their child’s answer about the degree of urinary incontinence.


*“[…] I now notice that she quickly responds that everything is fine. ‘I don’t have any accidents anymore. I’ve been dry for an entire week.’ Even though that is not the case. I think that is partly due to […] that she simply prefers not to answer such questions. […] and that this is the quickest way to stop the questions”.*
Mother P2

Other parents mentioned that their child got used to talking about it throughout the treatment and felt more at ease to discuss it.


*“I mainly think that just talking about it, makes it a subject that you can simply discuss with whoever is around at the moment and that it is not something you only discuss at home, so to speak”.*
Mother P4

Experiences with the group setting were diverse and included several aspects. Most parents indicated that merely seeing others with the same condition provided recognition and a feeling that they were all in this together, whilst one parent mentioned seeing another older child with LUTD made her worry about the future of her own child.


*“[…] then at least I’m one hundred percent sure I’m not the only one”.*
Child P3 (10 years)


*“And for me, as a mother, that was very comforting, that I thought ‘Ah, I see’ her son was 12 at that time, so they were a lot further along in the trajectory. And they only had the treatment for the nights left. So that’s when I thought ‘Ah, yes, there is a light at the end of the tunnel somewhere, and it can get better’”.*
Mother P10

Facilitating the treatment in a group setting and, therefore, seeing others at ease during treatment led to reassurance and helped the children to be less anxious.


*“That girl was just very relaxed and that made me feel a little more relaxed too”.*
Child P3 (10 years)


*“I appreciated not being all alone in a room and it was also kind of cozy”.*
Child P9 (9 years)

The interviews further revealed parents to be interested in peer contact.


*“And I have to say that I was also very curious about, well curious… yes, […] you think ‘Oh, maybe I can talk to someone once to get a little confirmation, a little recognition from each other”.*
Mother P10

Despite the mentioned interest in peer contact in the interviews, few actual examples of peer contact occurred during the individual PTNS sessions. Lack of time, inability to make contact as others were busy on their phone or feeling insecure to start a conversation were mentioned as possible explanations. One parent indicated that for her it was a barrier to start a conversation, as she was unsure whether it was appropriate to talk to someone else while not violating their privacy.


*“[…] recently there was a child who came for the very first time and who was clearly nervous and I could see the parents were apprehensive too, and for a minute I thought, maybe I should, well, you know we’re in this together and to just tell them not to worry and that it will be okay and I didn’t mention it because I thought, well, it kind of feels like an intrusion into their own affair”.*
Mother P5

Children themselves did not express a clear opinion about the peer contact. Several parents suggested a role for the healthcare provider to initiate more contact between parents and children.


*“Usually once you’re talking, the conversation runs its course. But perhaps they could also play a role when it comes to that. To try to loosen up the parents a little”.*
Mother P10

## 4. Discussion

In our group of 101 therapy refractory-children with LUTD, an overall improvement of urinary symptoms after PTNS was seen in 42%, with a complete response in 10%. Frequency voiding chart parameters—maximum and average voided volumes—increased as well throughout the treatment, as did disease-specific QOL in both parents and children. Improvement in QOL was independent of the actual effect of PTNS on urinary symptoms.

A large variation in response rates after PTNS treatment is seen, as in other studies, varying between 31–78% [11]. One randomized controlled study compared transcutaneous tibial nerve stimulation to a sham and reported a complete response rate as high as 71%, compared to 13% in the sham group [18]. Others report no differences in response rates and report improvement in 45–66% in both the stimulation and sham groups [19]. As most previous studies describe the outcome of transcutaneous-stimulation techniques using transdermal pads instead of percutaneous-stimulation techniques using needles—used in our study—comparison of their results to ours is difficult. In addition, differences between stimulation sites (posterior tibial nerve or sacral dermatomes), stimulation schedules (daily, twice a week, 3 times a week, once a week), study population (therapy-naive or therapy-refractory children), diagnosis (overactive bladder, dysfunctional voiding, neurogenic lower urinary tract dysfunction), definition of response and even stimulation parameters (amount of Hz, duration of individual session) further complicate this matter.

In our study, the average and maximum bladder capacity increased throughout PTNS treatment, which has been reported in other studies as well [11], but the clinical relevance of 25 mL and 13 mL, respectively, is to be questioned.

Overall, the PIN-Q scores of both parents and children decreased throughout PTNS treatment, implicating QOL to be less affected by LUTD after PTNS treatment than before. Interestingly, this improvement was also seen when patients had no actual response in urinary symptoms, which has also been described by other authors. In a randomized controlled trial by Jafarov et al., transcutaneous posterior tibial nerve stimulation was compared to a sham, and QOL significantly improved in both the stimulation and the sham group [20]. This suggests that other factors may be responsible for the improvement in QOL.

As PTNS is offered in a group setting in our clinic, one might expect peer contact to be one of the responsible factors influencing well-being. This was supported by our interview data, where recognition in seeing others with the same condition was reported repeatedly and positively valued by parents and children. Keeping in mind that the average child in our study had received 1.5 years of previous treatments, it is striking to see that PTNS was the first opportunity for them to see and meet peers. As LUTD has a significant impact on QOL, including diminished self-esteem, it appears important that children and parents understand that they are not the only ones [3,4]. Our results further reveal that merely offering a treatment in a group setting is not enough to initiate actual peer contact. Due to various reasons, contact with others hardly occurred in our group, despite the broad interest reported by subjects in the interviews. As LUTD appears to be a taboo topic, health-care providers should not only invest in creating possibilities for peer contact but also stimulate or initiate conversation among peers. This might provide both parents and children with coping strategies and possibly result in better acceptance of their condition.

Despite anxiousness about needle insertion, PTNS was well-tolerated and none of the children had to discontinue the treatment due to fear or pain. This is in line with the literature [9]. Time investment and logistics are known disadvantages of PTNS and self-administered home-based PTNS after in-hospital training or implantable devices are already available in the adult population [21,22]. However, for PTNS at home to be offered, its benefits should outweigh the valuable effect of seeing others with the same condition and the possibility for peer contact, in the case of treatment in a group setting. In addition, the regular time investment—as in weekly hospital visits—stimulates attention and awareness to one’s bladder and lower urinary tract and might be favorable given the relatively high placebo effect seen in patients treated with neuromodulation [19,23]. A combination of starting PTNS in a group setting and continuing home treatment after several sessions s might be a possible solution in the future.

To our knowledge, this is the first study reporting quantitative outcomes as well as qualitative outcomes of PTNS treatment in children and their parents. In addition to functional outcomes and validated QOL questionnaires scores, experiences with PTNS treatment were further explored by interviewing children and their parents. The retrospective character in the quantitative part of this study is a limitation, as is the fact that not all subjects filled in the PIN-Q questionnaire due to its implementation in a later stage. Interviews were conducted by independent research members for purposes of scientific integrity and to obtain an objective unbiased idea of how children and their parents experienced PTNS, as much as possible. However, if interviews would have been conducted by, for example, health-care providers familiar to the subjects, other results might have been found. Our interviews included children and their mothers but, unfortunately, no fathers. This could have influenced the results, as fathers’ experiences may differ from mothers’ experiences.

## 5. Conclusions

In conclusion, PTNS is a well-tolerated treatment, with an overall improvement in 42% in therapy-refractory children with LUTD. Disease-specific QOL improvement is seen during treatment, regardless of an actual effect on urinary symptoms, in both children and their parents. Facilitating PTNS in a group setting led to valuable feelings of recognition. This emphasizes the importance of stimulating peer contact in children with LUTD and their parents and helps in coping strategies in this often-stressful condition.

## Figures and Tables

**Figure 1 ijerph-19-09062-f001:**
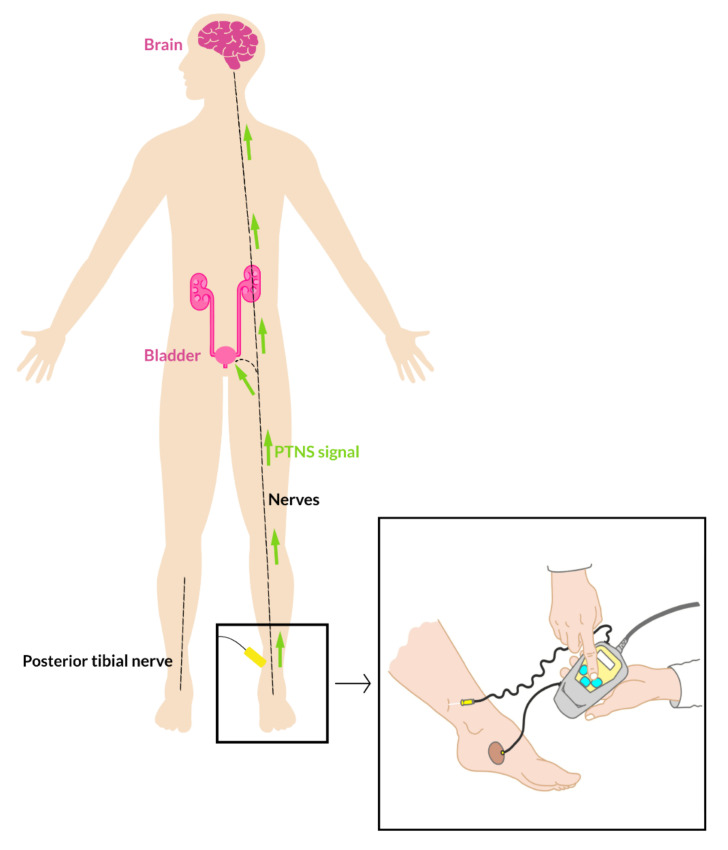
Schematic view of posterior tibial nerve stimulation.

**Table 1 ijerph-19-09062-t001:** Frequency voiding chart parameters and treatment response according to the ICCS *.

FVC **	Baseline	After PTNS	*p*-Value
Average voided volume in mL (Median/25th–75th Quartiles)	114 (77–150)	139 (100–175)	0.000
Maximum voided volume (Median/25th–75th Quartiles)	192 (130–240)	205 (150–250)	0.025
**Treatment response**			
Complete response (100% cure)	10%		
Partial response (50–99% improvement)	32%		
50–80% improvement		23%	
81–99% improvement		9%	
No response (0–49% improvement)	58%		

* ICCS International Children’s Continence Society; ** FVC frequency voiding chart parameters.

**Table 2 ijerph-19-09062-t002:** Disease-specific quality of life on the pediatric incontinence questionnaire (PIN-Q).

Total Score (Median/25th–75th Quartiles)	Baseline	After PTNS	*p*-Value
Child (N = 46)	25 (20–41)	19 (14–26)	0.001
Parent(s) (N = 46)	26 (18–38)	22 (11–30)	0.001

Higher scores indicate a lower quality of life.

**Table 3 ijerph-19-09062-t003:** Disease-specific quality of life on the pediatric incontinence questionnaire (PIN-Q), subdivided for treatment outcome (Median/25th–75th Quartiles).

	**No Response ***		**Partial Response ***		**Complete Response ***	
	T0	T1	T0	T1	T0	T1
Child	24 (19–24)	18 (14–18)	32 (22–44)	23 (16–32)	23 (16–37)	13 (6–13)
Parent(s)	23 (20–31)	22 (12–31)	22 (18–38)	22 (16–30)	36 (18–44)	22 (5–27)

* No response (0–49% improvement), partial response (49–99% improvement), complete response (100% cure); T0 = baseline, T1 = after PTNS.

**Table 4 ijerph-19-09062-t004:** Characteristics of the subjects interviewed (N = 11).

Characteristics		Child	Parent
Gender	Male	5 (45%)	-
	Female	6 (55%)	11 (100%)
Age (child)	6–8	4 (36%)	
	9–12	7 (64%)	
Treatment duration before PTNS (years)	0–3	6 (55%)	
	≥4 years	5 (45%)	
Response to PTNS *	NR	6 (55%)	
	PR/CR	5 (45%)	
Stage of PTNS and time of interview	At the start	2 (18%)	
	Halfway	4 (36%)	
	PTNS completed	5 (45%)	

* NR = no response, PR = partial response, CR = complete response.

## Data Availability

The datasets generated during and/or analysed during the study will be available upon request: L.L. de Wall, liesbeth.dewall@radboudumc.nl. These include anonymized data of participants, SPSS data and transcribed data in Atlas.ti. Data are available after publication and a data-transfer agreement and for academic purposes only.

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
