# Peer review of "Posterior Tibial Nerve Stimulation in Children with Lower Urinary Tract Dysfunction: A Mixed-Methods Analysis of Experiences, Quality of Life and Treatment Effect"

_ijerph, 2022, doi:10.3390/ijerph19159062_

Round 1

Reviewer 1 Report

In this manuscript, the authors used a mixed-methods analysis to gain insight into experiences of children treated with posterior tibial nerve stimulation (PTNS) and their parents, the effect of treatment on quality of life and effect of PTNS on urinary symptoms. The results obtained can help to establish optimal treatment protocols in the future. In the opinion of this reviewer, only a few points need to be addressed before the article is accepted for publication.

Minor comments:

Line 89: The authors should indicate the total number of children.

Lines 125-126: “…or those who had completed PTNS in the past 18 months as well as their parents, were invited”. Why in the past 18 months?

Line 141: “Interviews were conducted by an independent research member”. It may be helpful to mention this as a limitation, as this person may have asked questions differently than other potential interviewers, which could have biased the results.

Line 162: “Data of 101, treated” should be “Data of 101 children treated”.

Lines 172-174: “Both average voided volume and maximum voided volumes increased significantly throughout PTNS treatment, respectively 20 ml…”. Line 378: “respectively, 20 and…). However, in Table 1, the average voided volume baseline is 114 and after PTNS 139 (difference: 25 ml).

Lines 202-203: “A total of 12 parents were asked and 11 were willing to participate”. Why were not more parents included in this part of the study?

Table 1, Table 2 and Table 3: “Quartiles” should be “range”.

Reviewer 2 Report

In the study's objective, the authors should also describe the qualitative section of their study. 

In the method section please mention the inclusion/exclusion criteria.

Please mention the previous treatment method of the eligible children?

Did you limit the report to non-neurogenic cases? Please describe.

Round 2

Reviewer 2 Report

The authors responded to my comments satisfactorily. 

one minor comment:

Please replace the neurogenic bladder with neurogenic lower urinary tract dysfunction.